# Deltaic Burial of Authigenic Calcite Modulates the Carbon Balance of Hardwater Lakes

Benedict V. A. Mittelbach<sup>1,2</sup>, Margot E. White<sup>1,3</sup>, Timo M. Y. Rhyner<sup>1</sup>, Negar Haghipour<sup>1,4</sup>, Marie-Elodie Perga<sup>5</sup>, Nathalie Dubois<sup>1,2</sup>, Timothy I. Eglinton<sup>1</sup>

15 Correspondence to: Benedict V. A. Mittelbach (bmittelbach@ethz.ch)

Abstract. Inland waters play an important role in the terrestrial carbon cycle by burying carbon in aquatic sediments while simultaneously releasing CO<sub>2</sub> to the atmosphere and laterally exporting carbon along the land-ocean aquatic continuum. Especially in hardwater lakes, the close connection between primary production and calcite precipitation results in a poorly understood balance of carbon burial and release, with stronger coupling of organic and inorganic processes than in softwater lakes. To better understand these dynamics, we analyzed organic and inorganic carbon fluxes in a yearlong (June 2022 to June 2023) sediment trap study in Lake Geneva, the largest natural lake in Western Europe. Two sediment traps - one deployed in the subaqueous delta of the upper Rhône River, the other in the lake's deepest basin - were sampled monthly. Analyzing radiocarbon (<sup>14</sup>C) signatures of particulate organic and inorganic carbon allowed us to resolve allochthonous (external) and autochthonous (internal) contributions to absolute carbon fluxes. We found that the flux of autochthonous particulate inorganic carbon in the river-proximal deltaic site was approximately four times higher than in the distal one. This is likely the result of calcite precipitation driven by increased fluvial supply of nutrients and suspended carbonate-bearing particles. Sediment core analysis in the same location suggests efficient preservation of this calcite over centennial timescales, which we conservatively estimate around 7-10 Gg C yr<sup>-1</sup> lake-wide. This indicates at least partial offset of the CO<sub>2</sub> released during calcite precipitation and is an important flux to be considered in mechanistic carbon cycle models.

## 1 Introduction

Inland waters are key sites for integrating, processing, and storing carbon (C), thus representing a key component of the terrestrial C cycle. The long residence time of water in lakes and reservoirs makes them important sites for the processing of C, leading to its burial or outgassing (Cole et al., 2007; Tranvik et al., 2018). The burial of particulate organic carbon (POC)

<sup>&</sup>lt;sup>1</sup>Department of Earth and Planetary Sciences, ETH Zurich, Zurich, Switzerland

<sup>&</sup>lt;sup>2</sup>Department of Surface Waters Research and Management, Eawag, Dübendorf, Switzerland

<sup>&</sup>lt;sup>3</sup>Department of Earth, Ocean and Atmospheric Sciences, University of British Columbia, Vancouver, Canada,

<sup>10 &</sup>lt;sup>4</sup>Laboratory for Ion Beam Physics, Department of Physics, ETH Zurich, Zurich, Switzerland

<sup>&</sup>lt;sup>5</sup>Faculty of Geosciences and Environment, Institute of Earth Surface Dynamics, University of Lausanne, Lausanne, Switzerland

or particulate inorganic carbon (PIC) in lake sediments can act as a sink of C. However, at the same time, outgassing of CO<sub>2</sub> from the dissolved inorganic carbon (DIC) pool to the atmosphere, e.g., from the remineralization of organic carbon (OC), represents a source of CO<sub>2</sub>. The balance between burial and outgassing processes varies both spatially and seasonally. Further, implications for regional and global C budgets critically depend on the source and type of processed C.

Large lakes, in particular, capture significant amounts of organic and inorganic C from various sources in their catchments (allochthonous C) and from in-lake processes such as aquatic primary productivity (autochthonous C), with differing climatic implications. Allochthonous POC (POC<sub>Allo</sub>) can consist of fast-cycling C such as biospheric POC, which rapidly incorporates atmospheric CO<sub>2</sub>, or from soils, where it can be stored on up to millennial timescales, acting more subtly on present-day climate (Eglinton et al., 2021). The erosion and re-burial of POC stored in bedrock on even longer, geological timescales, termed petrogenic POC, does not influence present-day CO<sub>2</sub> levels. Similarly, allochthonous PIC (PIC<sub>Allo</sub>) contains eroded fragments of carbonate rocks, which formed millions of years ago, along with PIC precipitated in the fluvial system. Autochthonous POC (POC<sub>Auto</sub>) and PIC (PIC<sub>Auto</sub>) consist of biomass from primary productivity and carbonate particles from calcite precipitation respectively. Both types of C are sourced from the dissolved inorganic carbon (DIC) pool within the lake. Commonly, most of the DIC is atmospheric, but chemical weathering of carbonate rocks leads to the incorporation of rock-derived C into DIC (Blattmann et al., 2018; White et al., 2025). Therefore, it is important to precisely constrain the origin of DIC, as only the atmospheric fraction contributes to net atmospheric CO<sub>2</sub> drawdown of POC<sub>Auto</sub> and PIC<sub>Auto</sub>.

Several processes can influence CO<sub>2</sub> outgassing from the DIC pool to the atmosphere. In the past, lake metabolism, i.e., the balance between aquatic productivity and remineralization of OC, and pH-driven transformation of bicarbonate (HCO<sub>3</sub>) to CO<sub>2</sub> were seen as the main drivers of CO<sub>2</sub> evasion (Duarte and Prairie, 2005; Many et al., 2024). However, calcite precipitation (PIC<sub>Auto</sub>) is increasingly recognized as a key process inducing CO<sub>2</sub> outgassing (Marcé et al., 2015; Khan et al., 2022; Many et al., 2024). In lakes with alkalinity greater than 1 mM, which account for almost half of the lake surface area worldwide, alkalinity is the main source for outgassing CO<sub>2</sub> and consequently the process significantly influences the inland water C budget (Khan et al., 2022; Many et al., 2024). Calcite precipitation is triggered by oversaturation of lake water with respect to calcite, which can be due to warming, (re-)dissolution of calcite, a rise in pH resulting from CO<sub>2</sub> evasion through turbulence or CO<sub>2</sub> drawdown by photosynthesis, or often a combination of these processes (Khan et al., 2021). Calcite precipitation can occur at low background levels during the warm summer period or in short, distinct pulses, termed whiting events (Escoffier et al., 2022; Heine et al., 2017). The process leads to the formation of one mole of CO<sub>2</sub> for each mole of calcite formed, while consuming two moles of alkalinity. While the CO<sub>2</sub> released during calcite precipitation can be taken up by photosynthesis, calcite precipitation ultimately contributes to net CO<sub>2</sub> outgassing on an annual scale, by making the lake less of a CO<sub>2</sub> sink in summer (Many et al., 2024). The burial of calcite in sediments removes C from the active C cycle, acting as a potential sink of atmospheric CO<sub>2</sub>. The climatic implications are further modulated by the origin of the DIC pool.

55

Radiocarbon (<sup>14</sup>C) offers a valuable tool to constrain different sources of C by utilizing the content of <sup>14</sup>C in a sample to inform about the timing since a pool last exchanged with the atmospheric <sup>14</sup>C pool (Graven et al., 2022; Levin and Hesshaimer, 2000; Trumbore, 2009). Pools in active exchange with the atmosphere will incorporate atmospheric <sup>14</sup>C levels, while rock-derived

C is free of measurable <sup>14</sup>C given its ~5700-year half-life. Pre-aged C, such as OC from soils, presents an intermediate case in this regard. DIC produced from the weathering of carbonate rocks with atmospherically-derived carbonic acid (H<sub>2</sub>CO<sub>3</sub>) will incorporate atmospheric and rock-derived C at equal proportions (Blattmann et al., 2018)

In this study, we determine the <sup>14</sup>C characteristics of sinking PIC and POC from two sediment trap locations in Lake Geneva, a perialpine hardwater lake with well-documented calcite precipitation (Escoffier et al., 2023; Many et al., 2024) and considerable terrestrial C influx, the latter primarily from the upper Rhône River (Lane et al., 2019). The aim of this study is to analyze spatial and seasonal variations in the sources, fluxes and pathways of CO<sub>2</sub> drawdown. We compare these fluxes to those from a sediment core spanning the last ~100 years in order to assess the long-term fate of PIC and POC. We show that combined <sup>14</sup>C measurements of both PIC and POC can be a valuable tool to precisely quantify net atmospheric CO<sub>2</sub> sequestration, and to highlight the potential importance of these processes with respect to the carbon balance in large, hardwater lakes.

## 2. Materials and Methods

# 2.2 Study Area



Figure 1: (a): Geological map (swisstopo, 2005) of the catchment area of Lake Geneva and sampling sites mentioned in the text. Red square indicates extent of panel (b). (b): Bathymetry (Swissbathy3D, swisstopo) of the lake and position of sampling sites. (c): Schematic setup of the sediment traps used, depth information refers to sampling site CHL. Basemaps panels (a) and (b): ESRI

Lake Geneva (Fig. 1), located between the Jura mountains to the north and the Alps to the south, has a surface area of 580 km², a maximum depth of 310 m, and a volume of 89 km³, making it the largest natural lake in Western Europe. Lake Geneva is an oligo-monomictic perialpine lake, which only mixes in very cold winters, on average every 7 years (Gaudard et al., 2017). After a period of eutrophication that peaked in the late 1970s, the lake has reached a meso-oligotrophic status with a total phosphorus (TP) concentration below 17 μg L⁻¹ (Berthon et al., 2014; Perga et al., 2016). Lake water alkalinity ranges between 1.2 and 2 mM, making it a moderately hardwater lake and reflecting the carbonate rock-rich catchment (Escoffier et al., 2023). The Upper Rhône River (Fig. 1) provides about 75-80% of the lake's inflow (Cipel, 2020) and roughly 85% of the sediment

(Loizeau et al., 2012). The river is also the primary source of phosphorous (Perga et al., 2016) and plays a key role in the sediment distribution within the lake basin (Jaquet et al., 1983). Coarser, river-derived material settles relatively close to the inflow, forming a large subaqueous delta (Silva et al., 2019; Piton et al., 2022). Finer particles are transported farther into the lake, especially due to interflow dynamics (Ishiguro and Balvay, 2003; Piton et al., 2022).

The Upper Rhône catchment consists of three major lithological units (Fig. 1a). The External Massifs in the east contain mostly granitic, magmatic rocks, the Penninic Nappes in the south comprise metamorphic and ophiolitic rocks and the Helvetic Nappes in the north consist of carbonate rocks (Schmid et al., 2004). Accordingly, provenance analyses show that the sediments transported into the lake derive about 57% from the External Massifs, 23% from Penninic Nappes, and 20% from Helvetic nappes (Stutenbecker et al., 2018). Consequently, we expect deposition of PIC<sub>Allo</sub>, derived from carbonate rocks, as well as rock-derived POC, e.g., from sedimentary rocks in the Helvetic and Penninic nappes. Near the inflow, at Porte du Scex (Fig. 1(a)/(b)), a monitoring station operated by the Swiss Federal Office for the Environment (FOEN) continuously records environmental parameters of the Upper Rhône. Additionally, monitoring and research infrastructure at the *LéXPLORE* floating laboratory (Wüest et al., 2021) provide detailed insights into Lake Geneva's dynamics. This comprehensive observational network and diverse data availability make Lake Geneva an ideal environment to study C cycling processes and to apply natural isotopic tracers (esp. <sup>14</sup>C) to delineate C sources and fates.

## 2.2 Sample collection





Sediment traps (Fig. 1(c)) were deployed in two locations with contrasting expected sedimentation properties. Trap CHL was deployed in a river-proximal location (~100 m depth) to capture allochthonous material emanating from the upper Rhône River. Trap SHL2 was placed at a distal location in the deepest part of the basin (~300 m depth) to capture open water sedimentation and predominantly autochthonous material. The sediment traps consisted of 4 PVC tubes each with a length of 90 cm and a diameter of 9 cm that were placed 5 m above the lake floor to ensure sinking flux to be representative of flux to the sediment. Traps were deployed on a mooring comprised of buoys, an anchor and an acoustic releaser (Fig. 1(c)) and retrieved monthly from June 2022 to June 2023 and the contents emptied into PE containers that were kept frozen until further processing. Due to disturbance of the sediment surface while deploying the distal trap for the first time in June 2022, and loss of material while retrieving the same trap in April 2023 no data is available for these periods.

Sediment cores GEN22-03 and GEN22-04 were sampled from the same location (± 300 m due to drift while sampling) as that of the proximal sediment trap CHL. The sediment cores were acquired in October 2022 using a gravity core system (Eawag-63/S corer, Eawag, Switzerland) with a 63mm PVC core liner. After sampling, sediment cores were stored at +4°C until further processing. With the available equipment, it was not possible to collect sediment cores at the distal site.

## 2.3 Laboratory Analyses

Sediment trap material was freeze-dried using a Christ Alpha freeze-drier and weighed to compute the sedimentation mass flux. The age-depth relationship of GEN22-04 was determined by measuring the activity of <sup>137</sup>Cs in the sediment core from

gamma spectroscopy on freeze-dried, homogenized samples placed into plastic tubes at the Gamma laboratory at Eawag. We analyzed 28 of these samples using high-purity Germanium Well Detectors (Canberra or Princeton Gamma-Tech). The same material was used for subsequent analysis. Sediment density was assessed from 1cm<sup>3</sup> subsamples of the sections of GEN22-03, assuming equal core density as they were taken from the same locations and visually correlate well to one another.

We subsampled 76 sections of 1 cm thickness from GEN22-04, freeze-dried them, and weighed 2-5 g into PVC sampling tubes for bulk C analysis. We determined the organic and inorganic carbon content on 20-60 mg aliquots using a SoliTOC Cube Analyzer (Elementar GmbH, Langenselbold, Germany) at Eawag Dübendorf, determining total organic carbon (TOC) and total inorganic carbon (TIC) in weight %. We used a pure CaCO<sub>3</sub> standard and two standards with high and low TOC content (all provided by Säntis Analytical AG, Teufen, Switzerland) as reference materials. Standard reproducibility was better than 0.02%. For <sup>14</sup>C measurements of PIC (PI<sup>14</sup>C), we homogenized samples with a mortar and pestle to break up aggregates and placed 3-6 mg of sample aliquots into 12 ml exetainer vials. We evacuated the headspace with ultra-high purity He, then transformed PIC into CO<sub>2</sub> using 85% phosphoric acid (H<sub>3</sub>PO<sub>4</sub>) at 60°C for 1h. The released CO<sub>2</sub> was introduced into the Elemental-Analyzer-Accelerator-Mass-Spectrometer (EA-AMS) MICADAS system at the Laboratory for Ion Beam Physics (LIP), ETH Zürich (Synal et al., 2007) via the Gas-Interface-System (GIS) (Mcintyre et al., 2016; Ruff et al., 2010). C1 (IAEA) was used as blank for PI<sup>14</sup>C measurements, and C2 (IAEA) as a secondary reference material.

For the analysis of  $\Delta^{14}$ C and  $\delta^{13}$ C of POC (PO<sup>14</sup>C and PO<sup>13</sup>C), 10-40 mg of sample was transferred to silver capsules (Elemental Microanalysis, 8x8x15), and inorganic C was removed by vapor acidification with 12M HCl (Bao et al., 2019; Komada et al., 2008). After neutralization with NaOH, we wrapped the samples in tin boats (Elemental Microanalysis) and measured PO<sup>14</sup>C using the MICADAS system. The standards for <sup>14</sup>C were oxalic acid II (NIST SRM 4990C) and phthalic anhydride (Sigma, PN-320064-500g, LN-MKBH1376V). Radiocarbon measurements are reported as age corrected  $\Delta^{14}$ C values in per mil (‰) (Stenström et al., 2011)

Stable δ<sup>13</sup>C was measured on an EA-Isotope-Ratio-Mass-Spectrometer (EA-IRMS, precisION Isoprime, Elementar GmbH, Germany). As standards for <sup>13</sup>C atropine (Säntis, PN - SA990746B, LN- 51112), acetanilide (Merck, PN-100011, LN - K37102211229) and Peptone (Sigma, PN-P7750-100G, LN-SLBC5290V) were used and repeatability and intermediate precision of <sup>13</sup>C measurements were better than 0.16 ‰.

#### 2.4 Data Processing




Fluxes in the sediment traps were derived by weighing of the dried material. For the sediment core, we created a linear depth-density model, as well as a linear age-depth model from the measured <sup>137</sup>Cs peaks corresponding to 1963 (nuclear weapons tests) and 1986 (Chernobyl accident) and multiplied them into the mass accumulation rate (MAR). To identify PIC sources,

PIC<sub>Auto</sub> was assumed to derive from the lake's DIC pool, while PIC<sub>Allo</sub> was considered fluvially transported detrital carbonate rock devoid of measurable <sup>14</sup>C. The relative contribution of PIC<sub>Auto</sub> to total PIC flux was calculated using a two-endmember mixing model:

$$PI^{14}C_{obs} = \alpha \times PI^{14}C_{Auto} + \beta \times PI^{14}C_{Allo} \tag{1}$$

and





$$\alpha + \beta = 1 \tag{2}$$

PI<sup>14</sup>C<sub>Auto</sub> was based on DI<sup>14</sup>C data from White et al. (2025), with Δ<sup>14</sup>C values ranging from -201% to -152 %. PIC<sub>Allo</sub> from the Upper Rhône River was previously estimated to be purely rock-derived based on  $\delta^{13}$ C signatures (Aucour et al., 1999) and SEM microscopy of PIC (Escoffier et al., 2022). This assumption is further supported by <sup>14</sup>C analysis of three riverine PIC samples (Appendix A) which yielded virtually <sup>14</sup>C free results (Δ<sup>14</sup>C from -992‰ to -981‰). Consequently, PIC<sub>Allo</sub> was assumed to be radiocarbon dead (-1000 %). Monte Carlo simulations (n = 100,000) were performed, drawing DI<sup>14</sup>C from a flat distribution within the observed range, as these values bound an interval without clear justification for a Gaussian distribution, and PI<sup>14</sup>C from a normal distribution centered on the observed values with their measurement uncertainties. This approach estimated the relative contributions of PIC<sub>Auto</sub> and PIC<sub>Allo</sub> and their associated uncertainties. For POC, the approach is less straightforward, as POC<sub>Allo</sub> can display a range of <sup>14</sup>C values (Schwab et al., 2022; Rhyner et al., 2023). Here, we used the strategic placement of the two sediment traps to resolve POC inputs - on the assumption that the river-proximal trap captures allochthonous deposition and the distal one records largely autochthonous background flux. We used the POC flux in the distal trap as a baseline of autochthonous deposition and assigned the excess POC flux in the proximal trap to allochthonous inputs. We performed all data processing and descriptive statistical analysis using Python (3.10.9) with the NumPy (1.23.5) and SciPy (1.7.3) libraries. We reconstructed the total flux for allochthonous and autochthonous POC and PIC using the results of the end member assignment in combination with the mass flux for the sediment traps. As no source apportionment has been conducted for the sediment core samples, we compare these fluxes to the bulk accumulation rates of organic and inorganic C in the sediment.

## 3 Results



## 3.1 Mass and Carbon Fluxes

The total mass flux (Fig. 2) varied markedly between the two trap locations and seasonally. In the river-proximal trap, daily mass flux varies between 2.1 g m<sup>-2</sup> day<sup>-1</sup> in the December 2022 deployment period to 43.7 g m<sup>-2</sup> day<sup>-1</sup> in the August 2022 period, with an average annual flux of  $12.6 \pm 12.5$  g m<sup>-2</sup> day<sup>-1</sup>. In general, lower fluvial discharge months from November to April show much lower mass flux than the months from May to October. Mass flux in the distal trap followed similar seasonal trends but was an order of magnitude lower, ranging from 0.8 g m<sup>-2</sup> day<sup>-1</sup> in December 2022 to 3.4 g m<sup>-2</sup> day<sup>-1</sup> in June 2022, averaging  $1.8 \pm 0.8$  g m<sup>-2</sup> day<sup>-1</sup>.

Figure 2: (a): Surface (0-20 m) water temperature (red, solid line) recorded via CTD at the LéXPLORE platform (provided by Eawag/Datalakes) and discharge (blue, dashed line) of the Upper Rhône monitored at the Porte du Scex monitoring station (provided by FOEN). (b): Mass flux in g m<sup>-2</sup> day<sup>-1</sup> at the proximal site (CHL). Width of the bars indicates the duration of the deployment period. (c): Mass flux in g m<sup>-2</sup> day<sup>-1</sup> at the distal site (SHL2). The unusually high mass flux in the first deployment period at SHL2 is attributed to sediment resuspension associated with mooring deployment. Mass flux in April at SHL2 was not quantified due to a substantial loss of material after sampling. Note the different ranges for the y-axes in panels (b) and (c).

Compared to mass flux, the flux of POC (Fig. 3(a)) shows less pronounced contrasts between the two locations. POC flux in the proximal location varies from  $0.06 \text{ g m}^{-2} \text{ day}^{-1}$  in March 2023 to  $0.31 \text{ g m}^{-2} \text{ day}^{-1}$  in August 2022 (annual avg.,  $0.17 \pm 0.10 \text{ g m}^{-2} \text{ day}^{-1}$ ). In the distal trap, the lowest POC flux of  $0.04 \text{ g m}^{-2} \text{ day}^{-1}$  is observed in March 2023 and the maximum flux of  $0.21 \text{ g m}^{-2} \text{ day}^{-1}$  is reached in May 2023 (avg.,  $0.10 \pm 0.05 \text{ g m}^{-2} \text{ day}^{-1}$ ). POC flux is comparable between the two sites from November 2022 to May 2023. The proximal trap shows higher POC flux in months with high mass flux, i.e., August to October 2022 and June-July 2023. The proportionally higher POC flux compared to mass flux in the distal trap is the result of a higher

TOC content (5.46  $\pm$  1.65%) relative to the proximal trap (1.31  $\pm$  0.72 %), which we attribute to less pronounced dilution by allochthonous lithogenic material.

Figure 3: (a): Total flux of POC in g m<sup>-2</sup>day<sup>-1</sup>at the proximal (blue, dashed) and distal (orange, solid) site. (b): Total flux of PIC in g

m<sup>-2</sup>day<sup>-1</sup> at the proximal (blue, dashed) and distal (orange, solid) site.

PIC fluxes fall between 0.05 g m<sup>-2</sup> day<sup>-1</sup> (December 2022) and 0.53 g m<sup>-2</sup> day<sup>-1</sup> (May 2023) in the proximal trap and between 0.01 g m<sup>-2</sup> day<sup>-1</sup> (October 2022) and 0.15 g m<sup>-2</sup> day<sup>-1</sup> (June 2023) in the distal trap. On average, the PIC flux in the proximal trap  $(0.23 \pm 0.18 \text{ g m}^{-2} \text{ day}^{-1})$  is almost an order of magnitude higher than in the distal  $(0.04 \pm 0.04 \text{ g m}^{-2} \text{ day}^{-1})$ . This likely reflects the fact that lithogenic material contains allochthonous, detrital PIC from catchment erosion of carbonate rocks and is not, as with POC, and artifact of relative content (TIC [%]: proximal  $1.80 \pm 0.74\%$  vs. distal  $2.26 \pm 1.25\%$ ).

## 3.2 Carbon Isotopes and Pool-specific Fluxes



 $^{14}$ C was used to assess sources of C contributing to the sediment trap fluxes at the two locations: in the distal trap, PO $^{14}$ C ( $\Delta^{14}$ C values) vary from to  $^{-2}$ 14 ± 7 ‰ in March 2022 to  $^{-1}$ 79 ± 7 ‰ in June 2023, and average  $^{-1}$ 97 ± 12 ‰ (Fig. 4a). These values are in close agreement with DI $^{14}$ C determined for the water column (White et al. (2025), suggesting that POC supply to the distal trap is dominated by autochthonous inputs. Slightly lower  $\Delta^{14}$ C values of POC are reached in January to April 2023, which may reflect a higher relative contribution of allochthonous material during the colder, low-productivity period. Except for June 2023, PO $^{14}$ C values in the river-proximal trap are lower than lake DI $^{14}$ C and material in the distal trap (Fig. 4a). A detailed assessment of the  $^{14}$ C signatures of POC<sub>Allo</sub> is presented in Appendix A.

Figure 4: (a): Radiocarbon content of POC (Δ PO<sup>11</sup>C) in ‰. Blue circles indicate the proximal trap, green triangles the distal trap. (b): Radiocarbon content of PIC (Δ PI<sup>11</sup>C) in ‰. (c) Stable carbon isotope signature of POC (δ<sup>13</sup>C POC). For panel (a) & (b): The error bar in x-direction indicates the length of the deployment period. Error bars in y-direction the uncertainty of the <sup>11</sup>C measurement. The grey shaded area marks the range of values plus uncertainty observed at LéXPLORE by White et al. (2025). Grey circles mark individual observations at different water depths.

PI¹⁴C in the two traps shows much greater variability than PO¹⁴C (Fig. 4b). Δ¹⁴C values in the distal trap vary from -181 ± 7 ‰ in August 2022 to -828 ± 4 ‰ in January-February 2023 (avg., -380 ± 255 ‰). There is a clear divide into two groups: Δ¹⁴C values for the periods in November 2022 and January-February 2023 are very low (avg., -805 ± 28 ‰) while for the other periods they lie close to DI¹⁴C values (avg., -251 ± 77 ‰). PI¹⁴C in the proximal trap shows a similar pattern of variability but falls within a narrower range of Δ¹⁴C values (Fig. 4a), from -818 ‰ in April 2023 to -356 ‰ in December 2022 (avg., -257 ± 139 ‰). This least depleted period in December 2022, however, coincides with the lowest PIC flux in the time series, making the isotopic signal highly sensitive to even small variations in source contributions.

- Stable δ<sup>13</sup>C signatures of POC vary less strongly between the two traps, with the distal trap being slightly lower (avg. -28.7 ‰ ± 2.9 ‰) than the proximal trap (avg. -27.7 ‰ ± 2.2 ‰), likely reflecting a more autochthonous POC source at the distal site.

  Values in both traps increase from July to November 2022 before sharply decreasing, and lowest values are reached in summer 2023. In contrast to Δ<sup>14</sup>C, δ<sup>13</sup>C values are less diagnostic in resolving POC sources. The increasing values from July to November 2022 cannot be explained through DIC-concentration dependent fractionation, as DIC concentrations in the Lake increased during that period (White et al., 2025). Instead, changes in the composition of aquatic primary producers or increased microbial reworking due to longer residence times during convective mixing in Fall might be possible explanations.
- Assuming authigenic PI<sup>14</sup>C matches measured lake DI<sup>14</sup>C (White et al., 2025), while allochthonous PI<sup>14</sup>C is of rock-derived origin and free of measurable <sup>14</sup>C, we determined the relative contributions of authigenic versus external sources to sedimenting PIC at both locations. In general, the relative contribution of allochthonous PIC (PIC<sub>Allo</sub>) in the proximal trap (average 57% ± 17%) is higher than in the distal trap (24.56% ± 29.67%). The colder period from November to April has lower PIC fluxes and also shows the highest proportion of PIC<sub>Allo</sub> in the proximal (77.90% in April 2023) and distal (79.14% in January-February 2023) traps. The highest contribution of PIC<sub>Auto</sub> in the proximal trap occurred in December with 78.17%, but August 2022 and May 2023 also had high relative PIC<sub>Auto</sub> fluxes (~62%). The PIC accumulating in the distal trap during summer periods (August-September 2022, June-July 2023) is almost entirely autochthonous (94-99%).

By multiplying the contribution of PIC<sub>Auto</sub> with the total PIC flux, we are able to quantify the flux of PIC<sub>Auto</sub> for every observation period. The flux at the two sites is generally very similar, with PIC<sub>Auto</sub> flux at the proximal site only significantly exceeding that in the distal site in the summer periods of August and September 2022 and May and June 2023.





Figure 5: Modeled total annual fluxes of allochthonous and autochthonous organic and inorganic carbon for the proximal trap (a) and distal trap (b).

Over the entire observation period, overall carbon fluxes at the distal site were consistently lower than those at the proximal site (Fig. 5). At the distal site, total POC flux averaged around  $0.10 \pm 0.05$  g m<sup>-2</sup> day<sup>-1</sup> and total PIC flux around  $0.04 \pm 0.04$  g m<sup>-2</sup> day<sup>-1</sup>. While we assigned all of the POC to be POC<sub>Auto</sub> in our model, the <sup>14</sup>C-based model predicts that the majority of PIC is also autochthonous. Consequently, seasonal trends described for total fluxes of POC and PIC apply to the autochthonous fraction. In the warmer periods, i.e., August to October 2022 and May to July 2023, POC flux is highest and deposition of PIC<sub>Auto</sub> takes place, while no significant PIC accumulation is observed in winter. The total deposition at the distal location, excluding the June 2022 period (impacted by sediment disturbance) and the missing data from April 2023, is 31 g POC<sub>Auto</sub> m<sup>-2</sup>, 11.5 g PIC<sub>Auto</sub> m<sup>-2</sup> and 1.3 g PIC<sub>Allo</sub> m<sup>-2</sup>.

In contrast, the proximal site recorded significantly higher fluxes and more mixed contributions of autochthonous and allochthonous carbon sources. For example, in August 2022, the POC flux at CHL reached 0.31 g m<sup>-2</sup> day<sup>-1</sup>, with ~0.08 g m<sup>-2</sup> day<sup>-1</sup> derived from POC<sub>Auto</sub> and 0.23 g m<sup>-2</sup> day<sup>-1</sup> from POC<sub>Allo</sub>. Similarly, PIC fluxes at the proximal site were often more than an order of magnitude higher than at the distal site, peaking at 0.53 g m<sup>-2</sup> day<sup>-1</sup> in June 2023, with autochthonous and

allochthonous fractions frequently accounting for roughly equal proportions during high-flux periods. Deposition of PIC<sub>Allo</sub> occurs mostly during the high sediment flux summer months, but steady deposition of both PIC<sub>Auto</sub> and PIC<sub>Allo</sub> is observed even in winter. The total annual POC flux at the proximal site is 63.7 g m<sup>-2</sup>, comprising about 35.2 g POC<sub>Auto</sub> m<sup>-2</sup> and 28.6 g POC<sub>Allo</sub> m<sup>-2</sup>. Note that for deployment periods without corresponding distal POC data, all POC at the proximal location was assumed to be allochthonous, potentially resulting in a slight overestimation of the POC<sub>Allo</sub> contribution. The total annual PIC deposition is around 41.5 g PIC<sub>Auto</sub> m<sup>-2</sup> and 45.8 g PIC<sub>Allo</sub> m<sup>-2</sup>.

## 3.3 Sediment Core







The sediment core sampled at the river-proximal location (Fig. 6) showed three distinct lithotypes. The top 25 cm are characterized by alternating beige-grey and dark-grey layers of 0.5 to 1 cm thickness. From 25 to 40 cm the sediment continues to show regular variations in color, but the material is dark-grey to black. We interpret this interval to coincide with the eutrophication period in the lake which occurred in the 1960s to 1980s (Jenny et al., 2016). Sediment below 40 cm consists of alternating light grey and darker grey layers. Occasionally, small turbidite layers are identified from fining-upward intervals in the sediment, e.g., at 9, 51 and 67 cm depth. Further, some very organic matter-rich, black horizons appear at 10, 25, 30 and 66 cm. Density was measured from a core sampled in the same location and visually almost identical to GEN22-04, core GEN22-03. We observed that density (Fig. 6(c)) increases from 0.6 g cm<sup>-3</sup> in the top to 0.9 g m<sup>-3</sup> in the lower portion of the core. This could be the result of ongoing compaction of the sediment, or stem from the slight change in facies observed in the core, or from a combination of both factors. The content of TIC and TOC (Fig. 6d) follows similar trends. TOC content is fairly constant (~1 wt.%) throughout the core, with the exception of a few depths (10, 30, 50, 65cm) where TOC increases to values of 2-3 wt.%. TIC content is around 2 wt.% throughout the core but increases in the same intervals as TOC to values up to 5 wt.%. By multiplying the TOC and TIC content with the MAR we determined the accumulation rates (AR) of PIC and POC in the core (Fig. 6(e)). When comparing this to the total flux of POC in the proximal sediment trap, we find that the AR<sub>POC</sub> in the core is, in general, slightly lower than flux measured in the trap. However, in the section from 25 to 40 cm AR<sub>POC</sub> exceeds fluxes from the sediment trap, before it reduces again in the lower section of the core. The peaks visible in the AR<sub>POC</sub> are driven by the peaks in TOC content and should be interpreted with caution as our density model does not account for the presumably lower density of these organic-rich layers. In the top portion of the core, AR<sub>PIC</sub> follows similar patterns as AR<sub>POC</sub> being first slightly lower than in the trap and exceeding it in the 25-40 cm interval. In the lower portion however, AR<sub>PIC</sub> increases even more. The latter potentially reflects a higher content in detrital PIC in these visibly lighter facies. PI<sup>14</sup>C in the sediment (Fig. 6(f)) ranges from  $-946 \pm 1\%$  at 60 cm depth to  $-536 \pm 4\%$  at 25 cm. Values in the higher portion of the core agree with the mean PI14C signature of the proximal sediment trap, while values in the bottom section of the core are lower, consistent with a higher detrital contribution in this part. However, values are far from radiocarbon-dead, e.g.,  $-834 \pm 2\%$  at 75 cm depth, underlining the significant contribution of authigenic calcite even at depth.

Figure 6: (a): Line scan photograph of core GEN22-04, (b): Activity of <sup>137</sup>Cs in GEN22-04, two maxima in 23 and 39 cm depth are assigned to 1986 (Chernobyl disaster) and 1963 (nuclear weapons tests) respectively. (c): Dry bulk density of GEN22 -03 as determined with a volumetric sampler. A linear model is fit to the observed data, constant values assumed for depth above and below

measured data. (d): Content of TIC (blue) and TOC (green) in weight % in core GEN22-04, (e): Mass accumulation rate of PIC (blue) and POC (green) in GEN22-04. Uncertainty in shaded blue/green. Dashed lines indicate total annual deposition of PIC (blue) and POC (green) in the proximal sediment trap. (f): Radiocarbon content of PIC (\$\Delta\$ PI\$\$\subseteq\$ (\$\Delta\$ PI\$\$\subseteq\$ (\$\Delta\$ lines in the core GEN22-04 (blue dots). Shaded grey area shows mean and standard deviation of the average PI\$\$\subseteq\$ 10 fthe proximal sediment trap.

#### 4 Discussion




300

#### 4.1 Controls on POC Flux

Accumulation of POC in Lake Geneva reflects the complex interactions between allochthonous inputs from the Upper Rhône River and autochthonous production within the lake. This study sheds light on the spatial and seasonal variability in POC fluxes, driven by hydrological and biological processes, as well as the preservation dynamics observed in underlying sediments. Total mass flux as well as POC flux were consistently higher in the proximal compared to the distal trap, especially in the warmer months. We interpret this to be the result of increased fluvial discharge in spring and summer when meltwater influx increases, and rainfall events occur. Mass and POC flux in the proximal trap illustrate how increased Upper Rhône discharge delivers large quantities of terrestrial material and associated OC. In August and September 2022, POC flux in the proximal trap was nearly three times as high as in the distal trap, underlining the dominant role of fluvial input on POC flux during these periods. In contrast, POC fluxes in both traps were comparable during colder, low-discharge periods (November 2022 to April 2023) as a consequence of the reduced supply of fluvial material. These trends are also illustrated by PO $^{14}$ C.  $\Delta^{14}$ C values in the distal trap lie consistently within, or close to the observed range of DI $^{14}$ C, indicating a dominance of POC<sub>Auto</sub>, whereas the proximal trap exhibits lower  $\Delta^{14}$ C values that reflect input of pre-aged POC<sub>Allo</sub> from the catchment, especially during high-discharge periods. This interpretation is further supported by the  $\delta^{13}$ C values (< -28 % in most periods) in the

distal trap, characteristic of lacustrine organic matter (Randlett et al., 2015). Furthermore, the higher TOC content in the distal trap  $(5.46 \pm 1.65 \%)$  compared to the proximal trap  $(1.31 \pm 0.72 \%)$  reflects the less pronounced dilution by lithogenic material and the dominance of POC<sub>Auto</sub> at the distal site.

In addition to the riverine input, the relative contributions of POC<sub>Allo</sub> and POC<sub>Auto</sub> are influenced by in-lake primary productivity. During periods of high productivity, such as May 2023, as recorded by chlorophyll a monitoring at LéXPLORE (Datalakes/EAWAG), the distal trap recorded its highest POC flux, while the proximal trap continued to exhibit aged PO<sup>14</sup>C (lower  $\Delta^{14}$ C) signatures, suggesting concurrent inputs of autochthonous POC and terrestrial material. This complex interplay between external influx and internal productivity was apparent in June 2023. Despite similar  $\Delta^{14}$ C values of POC and DIC, the proximal trap recorded nearly twice the POC flux compared to the distal trap. This could be the result of higher autochthonous POC deposition in the proximal trap, influenced by nutrient delivery from the river (Cotte et al., 2023; Kiefer et al., 2015). Alternatively, it could be caused by the influx of POC<sub>Allo</sub> with a  $^{14}$ C signature similar to that of DI<sup>14</sup>C, which is possible, given the average PO<sup>14</sup>C of Swiss rivers was -164‰ in Summer 2021 (Rhyner et al., 2023). Further studies are required to resolve this question, but we note that POC<sub>Auto</sub> deposition in the proximal site may be slightly underestimated.

Insights into long-term POC burial are provided by sediment core GEN22-04. The top 25 cm of the core revealed lower AR<sub>POC</sub> compared to fluxes in the proximal trap, likely reflecting ongoing decomposition of POC following arrival at the sediment-water interface. During the eutrophic interval (25–40 cm depth), AR<sub>POC</sub> exceeded trap fluxes, coinciding with periods of heightened productivity (Loizeau and Dominik, 2005). In the deeper sections of the core (>40 cm), AR<sub>POC</sub> values decreased, approaching trap flux levels, suggesting effective preservation of POC, likely due to rapid burial and reduced remineralization in this high sedimentation deltaic site (Randlett et al., 2015; Sobek et al., 2009). The variability of TOC content (maxima of 2–3 wt.%), within the sediment core is primarily driven by short-term fluctuations that we interpret to be either linked to high-productivity events such as algal blooms or the influx of organic-rich terrestrial material. Concurrent increases in both TIC and TOC suggest a coupling between productivity and PIC deposition, as has been observed previously (Dittrich and Obst, 2004; Escoffier et al., 2023). However, it must be noted that our density model likely overestimates AR<sub>POC</sub> in these organic-rich layers due to the unaccounted lower sediment density.

#### 4.2 Controls on PIC Flux





PIC fluxes in Lake Geneva feature pronounced seasonal variability and contrasts between the proximal and distal locations. In general, PIC sedimentation is influenced by the interplay between allochthonous supply of detrital material and autochthonous calcite precipitation. The proximal trap consistently recorded higher PIC fluxes compared to the distal trap throughout all sampling periods, while seasonal variability, driven by hydrology and productivity, significantly influenced the relative contributions of PIC<sub>Auto</sub> and PIC<sub>Allo</sub>. The distal trap predominantly recorded PIC<sub>Auto</sub>, whereas the proximal trap exhibited significant contributions of PIC<sub>Allo</sub> material, especially during the high-discharge summer months.

At the distal site, PIC fluxes were almost entirely autochthonous, with PI<sup>14</sup>C values closely matching those of DI<sup>14</sup>C in most months. This pattern reflects the location, far from the riverine influence, and the dominance of calcite precipitation linked to in-lake primary productivity. However, during a few periods (e.g., November 2022 and January–February 2023), PI<sup>14</sup>C values indicated the co-deposition of allochthonous detrital carbonate. This increase in PIC<sub>Allo</sub> deposition could be linked to resuspension and lateral transport of (deltaic) sediment following enhanced bottom boundary layer turbulence during convective mixing in fall/winter (Fernández Castro et al., 2021). Given the overall low PIC flux in these months, even a small contribution of PIC<sub>Allo</sub> was sufficient to produce the observed <sup>14</sup>C-depleted signatures.

At the proximal site, PIC<sub>Allo</sub> deposition was more prevalent, especially during the summer, when high river-discharge led to high input of sediment, including detrital carbonates (PIC<sub>Allo</sub>). However, the proximal site also exhibited a higher flux of PIC<sub>Auto</sub> than the distal one. We identify two processes that may account for this contrast: (1) The shallower depth and thus shorter residence time in the water column at the proximal site might enhance preservation of PIC<sub>Auto</sub>. (2) The calcite precipitation in the proximal site could be higher than in the distal site. As calcite precipitation is closely tied to primary productivity in the lake (Many et al., 2024) and the river is the main source of nutrients (Kiefer et al., 2015; Cotte et al., 2023), the Upper Rhône discharge might fuel productivity in the proximal site, leading to increased PIC<sub>Auto</sub> precipitation. Further, suspended particles of detrital carbonate, which are more abundant in this location, have been identified as important nucleation sites for calcite precipitation in Lake Geneva (Escoffier et al., 2023), and are linked to occurrence of whiting events in the delta region (Escoffier et al., 2022).

Water-column dissolution of calcite particles in Swiss hardwater lakes is minimal, and dissolution rather occurs in the shallow sediment—water interface, driven by pH changes through organic matter decomposition (Müller et al., 2016; Müller et al., 2006). Consequently, our deep sediment traps provide accurate estimates of PIC flux to the sediment. Sediment core GEN22-04 then provides additional insights into long-term PIC accumulation and preservation. In the upper core section, AR<sub>PIC</sub> values were slightly lower than fluxes observed in the proximal trap, suggesting ongoing dissolution of PIC, although we cannot rule out potentially higher PIC flux in the studied year compared to previous ones. However, the eutrophic interval (approximately 25–40 cm depth) exhibited increased PIC deposition, consistent with the enhanced primary productivity observed during that period. This pattern highlights the link between calcite precipitation and biological productivity. PI<sup>14</sup>C in the top section of the core as well as during the eutrophic period agree with the mean values recorded by our sediment trap, suggesting ongoing deposition and preservation of the authigenic calcite. Importantly, the eutrophic interval coincides with the manifestation of the atmospheric bomb spike in the DI<sup>14</sup>C (Mittelbach et al., 2025) and slightly elevated values do not necessarily reflect more authigenic PIC. In the lower portion of the core (below 40 cm), non radiocarbon-dead PI<sup>14</sup>C signatures show a persistent flux of authigenic calcite, despite changes in sediment facies indicating a shift in depositional conditions. This interval, deposited before approximately 1960, predates significant hydrological interventions in the Upper Rhône catchment, including dam construction and channelization (Lane et al., 2019). We interpret this section to be dominated by detrital PIC, reflecting greater

delivery of carbonate-rich sediments from the catchment rather than increased PIC<sub>Auto</sub> precipitation. The  $\Delta$  PI<sup>14</sup>C of  $-834 \pm 2\%$  in the deepest sample would indicate ~20% of the PIC flux being authigenic assuming DI<sup>14</sup>C to be similar as today. In summary, despite the evidence for carbonate dissolution in the upper core, our findings suggest good preservation of autochthonous PIC in the deltaic site overall. This observation contrasts with modeling studies (e.g., Many et al., 2024) that predict complete dissolution of PIC<sub>Auto</sub> based on the lake's Ca<sup>2+</sup> budget. At least in the deltaic region, significant fractions of autochthonous calcite appear to be retained in the sediment, underlining the importance of site-specific conditions in determining PIC preservation.

## 4.3 Ramifications for the freshwater C sink






Using our point observations, we can derive first-order estimates of carbon fluxes for the different pools across the lake. PIC<sub>Auto</sub> flux ranged from approximately 40 g m<sup>-2</sup> yr<sup>-1</sup> in the proximal site to 12 g m<sup>-2</sup> yr<sup>-1</sup> in the distal site, corresponding to an extrapolated 7–30 Gg C yr<sup>-1</sup> if extended over the entire lake. The lower bound of ~7 Gg C yr<sup>-1</sup> should be considered conservative as it is based on the observations from the deepest site of the lake and calcite burial tends to be higher in shallower waters. Assuming that the deltaic sedimentation zone, with higher authigenic calcite flux, accounts for roughly 15% of the lake (Jaquet et al., 1983) yields a more realistic basin-wide estimate of about 10 Gg C yr<sup>-1</sup>. By comparison, ~17 Gg of calcite is modeled to precipitate annually (Many et al., 2024), indicating relatively effective burial of these particles. Although the long-term fate of PIC<sub>Auto</sub> in most parts of the lake remains unknown, our sediment core data indicates that the 3.5 Gg yr<sup>-1</sup> deposited in the deltaic zone are likely to be preserved. Radiocarbon analyses allow us to distinguish this from the 6.5 (3–25) Gg C yr<sup>-1</sup> of PIC<sub>Allo</sub> deposited in the lake.

Projecting POC<sub>Auto</sub> burial from the distal site to the entire lake suggests 20 Gg C yr<sup>-1</sup>, of which ~3 Gg C yr<sup>-1</sup> are buried in the delta. Current estimates indicate that ~8% of the NEP (8 Gg C yr<sup>-1</sup>) are buried annually (Steinsberger et al., 2021). Coupled with the ongoing POC decomposition in the sediment (Steinsberger et al., 2021), this implies that the distal trap's flux may not represent the final burial flux. Our data indicate that POC<sub>Allo</sub> accumulation in the delta sums to ~16 Gg C yr<sup>-1</sup>, consistent with an estimated 14 Gg C yr<sup>-1</sup> of Rhône-derived TOC input in 1974–2010 (Rodriguez-Murillo and Filella, 2015). This agreement suggests that our chosen proximal site is broadly representative of deltaic burial fluxes. Preservation of POC<sub>Allo</sub> in deltaic sediments is known to be efficient (Mittelbach et al., 2025; Sobek et al., 2009; Randlett et al., 2015). Overall, these findings illustrate that, despite their limited spatial coverage, data from our sediment trap deployments provide robust estimates of depositional fluxes for individual carbon pools in Lake Geneva.

## 410 5 Conclusions

By analyzing POC and PIC sources in two sediment traps in Lake Geneva, we were able to show that calcite precipitation can be an important contribution to C sequestration in this and other hardwater lakes. <sup>14</sup>C measurements enabled us to distinguish between external and internal PIC. In the subaqueous delta region, the observed authigenic PIC fluxes were four times higher

than in the distal sediment trap. Furthermore, analysis of a sediment core from the delta revealed that this authigenic PIC is at least partially preserved in the sediment. This challenges model predictions of complete dissolution and highlights the importance of calcite precipitation in hardwater lakes not only as a driver of CO<sub>2</sub> outgassing, but also as an under-recognized burial flux of carbon. Although sediment traps can only provide localized, short-term data, measured C fluxes are in agreement with existing, independent observations, and with data from longer-term sediment archives. Building on this work, future research should include broader spatial coverage to refine estimates for the entire lake, more detailed sampling of sediment cores to analyze the long-term fate of deposited POC and PIC in Lake Geneva, and extension to other hardwater lake systems.

## Appendix A: Radiocarbon Signatures of Allochthonous PIC and POC

By comparing the different PO<sup>14</sup>C signatures between the two traps, we estimated the PO<sup>14</sup>C<sub>Allo</sub>. To avoid unrealistic results (i.e., very low PO<sup>14</sup>C<sub>Allo</sub>), we only applied this approach for months where the POC flux of the proximal trap exceeded that of the distal trap by at least 20%. This criterion yielded seven valid periods (August to November 2022; January to March, and June and July 2023). We applied Monte-Carlo simulations (n = 100,000) to account for uncertainties in the <sup>14</sup>C measurements. By using the POC flux in the distal trap as an autochthonous baseline and assigning the excess POC flux in the river-proximal trap as allochthonous, we were able to calculate the PO<sup>14</sup>C signature of this allochthonous fraction. The highest  $\Delta^{14}$ C value of POC<sub>Allo</sub> was reached in June 2023 with -189 ‰ ± 5 ‰, while the lowest PO<sup>14</sup>C<sub>Allo</sub> signature occurs in the November 2022 with -491 ‰ ± 21 ‰ (Fig. A1). The average PO<sup>14</sup>C<sub>Allo</sub> is-341 ‰ ± 103 ‰, or -294 ‰ ± 48 ‰ when using an average weighted for individual uncertainty. These values are in agreement with observations for riverine suspended PO<sup>14</sup>C by Rhyner et al. (2023) who reported -218‰ ± 7‰ for Summer 2021 and -369 ± 7‰ for Summer 2022; the latter was measured as part of this study using the sampling and measuring protocol outlined in Rhyner et al. (2023). Increased Upper Rhône discharge appears to

coincide with younger POC<sub>Allo</sub>, as has been observed in other Swiss mountainous rivers (Gies et al., 2022; Schwab et al., 2022), but no statistically significant relationship emerges. The March 2023 period deviated from this trend, illustrating the inherent variability of riverine PO<sup>14</sup>C. These large variations within a single year underscore the need for long-term, high-resolution monitoring to accurately constrain fluvial PO<sup>14</sup>C sources and export.

Figure A1: Modeled <sup>14</sup>C signature of allochthonous POC in the proximal trap for periods with POC flux more than 20% higher than in the distal trap. Grey are shows standard deviation as determined by Monte-Carlo simulations. Dashed blue line shows discharge data of the Rhône River at Porte du Scex provided by the Swiss Federal Office for the Environment (FOEN). Shaded orange zone shows range of measures upper Rhône PO<sup>14</sup>C by Rhyner et al. (2023).

To better constrain the riverine PIC endmember, we analyzed the PI<sup>14</sup>C signature of three Rhône River suspended sediment samples, collected ant the Porte du Scex NADUF station in Summer 2023, Spring 2023 and Summer 2021, using the protocol described by (Rhyner et al., 2023). All samples gave virtually <sup>14</sup>C free results with  $\Delta^{14}$ C of –992 ± 1 ‰ (July 2023), –985 ± 1 ‰ (April 2023), and –981 ± 1 ‰ (July 2021).

## Acknowledgments






We thank Sébastien Lavanchy and Guillaume Cunillera (EPFL/LéXPLORE) and Anita Schlatter (Eawag) for organizing and facilitating field sampling. We acknowledge the technical and administrative support of the entire LéXPLORE team and the five LéXPLORE partner institutions: Eawag, EPFL, University of Geneva, University of Lausanne, and CARRTEL (INRAE–USMB). We are grateful to Alexander Brunmayr, Sarah Paradis, Remo Röthlin, Madeleine Santos, Charlotte Schnepper, and Aline Wildberger for field assistance. We thank Pascal Rünzi (Eawag) for gamma spectroscopy and Marcel Walter (Eawag) for SoliTOC measurements. AMS measurements were supported by the Laboratory of Ion Beam Physics at ETH Zurich. We also thank all contributors to the Radiocarbon Inventories of Switzerland project. We used ChatGPT (OpenAI) and Grammarly (Grammarly Inc.) to enhance writing clarity and assist with Python coding. This work was funded by the Swiss National Science Foundation Sinergia Grant No. 193770.

## **Author Contributions**

BVAM led the conceptualization and design of the study with contributions from TIE and ND. Method development was carried out by BVAM, MEW, and NH. BVAM implemented the software and performed data curation and formal analysis with input from MEW, MEP, and NH. Field investigation was conducted by BVAM and MEW. Resources were provided by TIE and ND. BVAM wrote the original draft of the manuscript with contributions from TIE, MEW, MEP, and ND. All authors contributed to review and editing. Visualization was prepared by BVAM. Project supervision and administration were carried out by ND and TIE, who also acquired funding for the study.

## **Data Availability**

All data presented in this study as well as code required to reproduce the results is on Zenodo at doi.org/10.5281/zenodo.15102592. All data from the ongoing monitoring at the LéXPLORE platform is openly available at www.datalakes-eawag.ch.

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
