# Peer review of "Deltaic Burial of Authigenic Calcite Modulates the Carbon Balance of Hardwater Lakes"

_EGUsphere, 2025_

## Author Comment (AC1)

We sincerely thank the reviewer for the very positive assessment and the helpful comments, which improved the clarity of the manuscript.

In addition to addressing the specific reviewer comments, we have incorporated new PI$^{14}$C measurements from the Rhône River (Appendix A) and sediment core GEN22-04 (now included in the Results, Section 3.3, and discussed in Section 4.2). These data provide further constraints on long-term PIC preservation and complement the trap-based flux observations.

Below we address each point in turn and indicate the corresponding changes made to the manuscript.

L138. Should this be C2?

We thank the reviewer for catching this and corrected C" to C2.

L164. Is there an advantage to using a flat distribution for the source end member (DIC) and a normal distribution (PIC) for the samples in the Monte Carlo simulation? If not, a sentence on the justification of this approach here would be useful.

For the DIC endmember, we applied a flat distribution to reflect the observed range across multiple measurements and depths. Because the values span a range without a clear justification for representing them as a mean with standard deviation, we chose to sample uniformly across the entire observed range. This avoids imposing an artificial Gaussian structure that is not evident. In contrast, a normal distribution was applied for PIC samples, as these are based on single measurements with well-defined, normally distributed analytical uncertainties. We have adjusted line 164 to read:

"Monte Carlo simulations (n = 100,000) were performed, drawing DI$^{14}$C from a flat distribution within the observed range, **as these values bound an interval without clear justification for a Gaussian distribution**, and PI$^{14}$C from a normal distribution centered on the observed values with their measurement uncertainties."

Section 4.3. Are the values calculated here just scaled based on the surface area of the lake? I appreciate that this is delineated as a first order estimate, but is a simple areal scaling of these values appropriate? Some clarification/justification here would be appreciated to add a bit more veracity to this section, which is important to clarify the importance of the study's findings.

We thank the reviewer for pointing out the need to clarify the assumptions of our lake-wide upscaling. We have now emphasized that the extrapolations made are based on areal scaling and should be regarded as order-of-magnitude estimates. We further highlight that the lower bound is conservative, since calcite burial tends to be higher in shallower areas. Lastly, we note that the proximal trap appears representative for the delta region because the upscaled allo. POC flux agrees well with independently measured Rhone input estimates. These clarifications have been added to Section 4.3.

---

## Author Comment (AC2)

We sincerely thank the reviewer for the detailed and constructive feedback, and for the very encouraging assessment of our work.

In addition to addressing the specific reviewer comments, we have incorporated new PI$^{14}$C measurements from the Rhône River (Appendix A) and sediment core GEN22-04 (now included in the Results, Section 3.3, and discussed in Section 4.2). These data provide further constraints on long-term PIC preservation and complement the trap-based flux observations.

Below, we address the reviewer's specific comments point by point and describe the corresponding revisions made.

Major Comments

Line 121: Were dissolution rates of settling calcite crystals considered? If estimable, could this affect the study's conclusions?

We thank the reviewer for raising this important point. Previous research in Swiss hardwater lakes has shown that dissolution of authigenic calcite particles in the water column is minimal, owing to their rapid settling velocity and the generally high DIC saturation (Müller et al., 2006; Müller et al., 2015). The sediment traps therefore provide a good estimate of the PIC flux to the sediment–water interface, where dissolution is more likely to occur. The effect of this is further investigated with the sediment core.

We have included lines 355 onwards:

*Water-column dissolution of calcite particles in Swiss hardwater lakes is minimal, and dissolution rather occurs in the shallow sediment−water interface, driven by pH changes through organic matter decomposition (Müller et al., 2016; Müller et al., 2006). Consequently, our deep sediment traps provide accurate estimates of PIC flux to the sediment.*

Line 154: The assumption that "PICAuto was assumed to derive from the lake's DIC pool" excludes any calcite formation in the Rhône or other tributaries. Please clarify why this possibility was ruled out.

Previously, this assumption was mainly based on existing research, such as the study by Escoffier et al. (2022), who performed SEM microscopy of calcite particles from the Rhone River and found no evidence for authigenic precipitation. However, we were now also able to analyze three samples of suspended Rhone River sediment for the 14C content of PIC. We found that Rhone PIC was virtually devoid of 14C, close to the detection limit, with signatures of –992 ± 1 ‰ (07/23), –985 ± 1 ‰ (04/23), and –981 ± 1 ‰ (07/21). We could thus successfully support our assumption that PIC$_{Auto}$ derives from in-lake DIC precipitation. We have added a clarifying sentence in the Methods (line 161 onwards)

*PICAllo from the Upper Rhône River was previously estimated to be purely detrital based on δ13C signatures (Aucour et al., 1999) and SEM microscopy of PIC (Escoffier et al., 2022). This assumption is further supported by 14C analysis of three riverine PIC samples (Appendix A) which yielded virtually 14C free results.*

and provide further details in Appendix A (Lines 425 onwards).

*To better constrain the riverine PIC endmember, we analyzed the PI14C signature of three Rhône River suspended sediment samples, collected ant the Porte du Scex NADUF station*

*in Summer 2023, Spring 2023 and Summer 2021, using the protocol described by (Rhyner et al., 2023). All samples gave virtually 14C free results with Δ14C of −992 ± 1 ‰ (July 2023), −985 ± 1 ‰ (April 2023), and −981 ± 1 ‰ (July 2021).*

Lines 168–169: It appears the potential lateral transport of allochthonous material from the Rhône to the distal trap was not accounted for. How can we be confident that a significant portion of allochthonous POC is not laterally transported? Given the overlap between riverine particle input and peak primary productivity, please justify this methodological assumption and discuss any potential implications for the results and their interpretation.

We thank the reviewer for this comment. We provide three arguments for why the lateral transport of Rhone-derived POC to the distal trap is not significant: (1) PO14C in the distal trap matches very closely the DI14C and is much less depleted than in the proximal trap, which is influenced by depleted riverine input. The short winter period with slightly lower PO14C coincides with minimal flux in the distal trap and is thus not quantitatively important. (2) Upscaling the proximal POC$_{Allo}$ flux to the riverine load reproduces Rhone-derived POC delivery reasonably well, suggesting that no major deposition is missed. (3) δ13C values at the distal trap are generally below −28‰, typical of lake aquatic organic matter (Randlett et al., 2015), except for October–December 2022. These less depleted values could be explained by increased remineralization after longer water column residence during convective mixing and do not indicate a systematic riverine contribution. We have added line 311:

*This interpretation is further supported by the δ13C values (< −28 in most periods) in the distal trap, characteristic of lacustrine organic matter (Randlett et al., 2015).*

Generally, a significant deposition of POC$_{Allo}$ (which we rule out based on the evidence presented), would lead to an underestimation of the POC$_{Allo}$ deposition not only in the distal trap, where it is not considered at all, but also in the proximal trap, where the distal flux is used as endmember for the POC$_{Auto}$ flux. This would result in an even higher POC$_{Allo}$ flux in the upscaling and a slightly younger average 14C signature of that POC$_{Allo}$.

Figure 4: Could the authors explain the abrupt increase in ΔPI14C observed in December 2022?

We thank the reviewer for highlighting this observation. We note that this period corresponds to the lowest PIC flux observed in the time series at both sites. Under these conditions, the PI14C signal becomes highly sensitive to even small variations in the relative contributions of authigenic and detrital PIC. We interpret the increase as a flux-driven effect: with very low overall PIC deposition, even a moderate authigenic component is enough to impact the isotopic signature strongly. L229 now reads:

*This least depleted period in December 2022, however, coincides with the lowest PIC flux in the time series, making the isotopic signal highly sensitive to even small variations in source contributions.*

Minor Comments

Line 300: Please specify "calcite precipitation events" to avoid confusion with meteorological precipitation.

We thank the reviewer for pointing this out, as the line in fact does relate to meteorological precipitation. We have changed the wording to "rainfall events".

Line 311: Regarding "such as May 2023," is there supporting data or a reference confirming elevated primary productivity during this period?

Chlorophyll A concentrations at the LéXPLORE platform show a strong increase, indicative of a phytoplankton bloom during that time, available at: www.datalakes-eawag.ch/datadetail/666

We have added:

*,as recorded by Chlorophyll A monitoring at LéXPLORE (Datalakes/EAWAG),*

to line 311.

Further, we have provided the link to the Datalakes portal in the data availability statement.

References

Escoffier, N., Perolo, P., Lambert, T., Rüegg, J., Odermatt, D., Adatte, T., Vennemann, T. and Perga, M.E., 2022. Whiting events in a large peri-alpine lake: Evidence of a catchment-scale process. *Journal of Geophysical Research: Biogeosciences*, *127*(4), p.e2022JG006823.

Müller, B., Meyer, J.S. and Gächter, R., 2016. Alkalinity regulation in calcium carbonate-buffered lakes. *Limnology and Oceanography*, *61*(1), pp.341-352

Müller, B., Wang, Y. and Wehrli, B., 2006. Cycling of calcite in hard water lakes of different trophic states. *Limnology and Oceanography*, *51*(4), pp.1678-1688.

Randlett, M.E., Sollberger, S., Del Sontro, T., Müller, B., Corella, J.P., Wehrli, B. and Schubert, C.J., 2015. Mineralization pathways of organic matter deposited in a river–lake transition of the Rhone River Delta, Lake Geneva. *Environmental Science: Processes & Impacts*, *17*(2), pp.370-380.

---

## Author Response (AR2)

Dear authors,

Thank you for preparing this revision. I gave your revised manuscript a last read and ask you to clarify the following minor issues. I regard this as a request for minor revision. Numbers refer to line numbers in the track-changes revision:

Dear Dr. Singer,

We thank you for taking the time to re-read our manuscript and for the very good points raised. We believe this substantially improved the manuscript further. Below we reply to the individual comments:

**19: I think this is written in a misleading and too simplified manner. The increased primary productivity will need a lot of CO2, thus limiting the CO2 outgassing. Please change and/or explain. Also see comment below.**

Thank you for pointing this out. The complex interplay of DIC, PIC and outgassing is central to our manuscript and we want to make sure the reader fully captures it. We have edited the line in the abstract to read:

Especially in hardwater lakes, the close connection between primary production and calcite precipitation results in a poorly understood balance of carbon burial and release, with stronger coupling of organic and inorganic processes than in softwater lakes.

We then provide more details in the introduction (see reply below).

**45: guess this needs a comma after "millions of years ago" to make sense.**

Thank you for pointing this out, we have added a comma to convey the intended message.

**49: I see no causality. Why would it be important? For what?**

While the incorporation of atmospheric  $CO_2$  into particles and its subsequent drawdown represents a direct drawdown of atmospheric  $CO_2$ , the dissolution and re-precipitation of carbonate rocks has no net effect on atmospheric  $CO_2$  levels. Assuming 100% of DIC are atmospherically derived would lead to an overestimation of the atmospheric  $CO_2$  sink of autochthonous particulate carbon. We have edited the sentence to make more clear that we are talking about the net atmospheric  $CO_2$  drawdown:

Therefore, it is important to precisely constrain the origin of DIC, as only the atmospheric fraction contributes to net atmospheric CO2 drawdown of POCAuto and PICAuto.

**59-64: While I agree that warming will drive CO2 from DIC into the atmosphere, I do think that such a simplified pathway from alkalinity to the atmosphere is not justified if calcite precipitation is driven by photosynthesis-induced increases in pH. When lake whiting is induced by blooming phytoplankton, the freed CO2 will be consumed in photosynthesis rather than lost in evasion, in my opinion. I ask the authors for a more nuanced phrasing of this text passage here and of the single sentence in the abstract. There may be alternative argumentation as well, but the current text seems a bit simple.**

Thank you again for this important clarification. We agree that our original text oversimplified the relationship between calcite precipitation (CP) and CO2 outgassing. It is correct that in summer, when the lake is undersaturated or close to CO2 saturation, CO2 released by CP is not directly outgassed but rather taken up by photosynthesis. However, CP makes the lake

less of a carbon sink during summer and, through this buffering effect, ultimately shifts to net annual CO2 outgassing (Many et al., 2024). The text has been edited to read:

While the CO2 released during calcite precipitation can be taken up by photosynthesis, calcite precipitation ultimately contributes to net CO2 outgassing on an annual scale, by making the lake less of a CO2 sink in summer (Many et al., 2024).

We attach the figure below from Many et al. (2024), which shows seasonal (winter to fall) and annual CO2 fluxes in Lake Geneva. The model with CP (orange) better matches observations (black) showing net annual CO2 emissions, while without CP (blue) the lake would be a net carbon sink, particularly during summer.

**74-77: Is it correct to use the term "CO2 drawdown" in these lines? Aren't you referring to the PIC and POC flux to the sediment and not to the decrease in CO2 concentration in the water?**

We agree that the phrasing of this sentence, similar to L49, was ambiguous. We are again referring to the net burial of atmospherically derived carbon and have edited the sentence for clarity as following:

We show that combined 14C measurements of both PIC and POC can be a valuable tool to precisely quantify net atmospheric CO2 sequestration, and to highlight the potential importance of these processes with respect to the carbon balance in large, hardwater lakes.

**163: does detrital here mean rock-derived? Could also be read as derived from modern organic detritus.**

Thank you for catching this ambiguity. We agree the readership of *Biogeosciences* might rather think of organic detritus. We have changed the phrasing to "rock-derived".

**353: could you get a bit more into mechanisms of this do-deposition, perhaps with help of a reference?**

Thank you for pointing out this opportunity to provide more detail. As riverine discharge is very low during these periods, we hypothesize that convective lake mixing is responsible for the redistribution of material. We have included the sentence:

This increase in PICAllo deposition could be linked to resuspension and lateral transport of (deltaic) sediment following enhanced bottom boundary layer turbulence during convective mixing in fall/winter (Fernández Castro et al., 2021).

We note that absolute PIC flux in these periods is vanishingly small at the distal location, and these redistribution processes, if of comparable magnitude for POC and PIC, have virtually no influence on POC deposition during this period.

**References**

Fernández Castro, B., Bouffard, D., Troy, C., Ulloa, H.N., Piccolroaz, S., Sepúlveda Steiner, O., Chmiel, H.E., Serra Moncadas, L., Lavanchy, S. and Wüest, A., 2021. Seasonality modulates wind-driven mixing pathways in a large lake. *Communications Earth & Environment*, *2*(1), p.215.

Many, G., Escoffier, N., Perolo, P., Bärenbold, F., Bouffard, D. and Perga, M.E., 2024. Calcite precipitation: The forgotten piece of lakes' carbon cycle. *Science Advances*, *10*(44), p.eado5924.